# Morphological Assessment of Nasopalatine Canal Using Cone Beam Computed Tomography: A Retrospective Study of 124 Consecutive Patients

**DOI:** 10.3390/diagnostics13101787

**Published:** 2023-05-18

**Authors:** Emmanouil Chatzipetros, Kostas Tsiklakis, Catherine Donta, Spyros Damaskos, Christos Angelopoulos

**Affiliations:** Department of Oral Diagnosis and Radiology, Faculty of Dentistry, National and Kapodistrian University of Athens, 2 Thivon Str., Goudi, 115 27 Athens, Greece; e.chatzipetros@gmail.com (E.C.); ktsiklak@dent.uoa.gr (K.T.); edonta@dent.uoa.gr (C.D.); sdamask@dent.uoa.gr (S.D.)

**Keywords:** cone beam computed tomography, dental implants, incisive foramen, maxilla, nasopalatine canal

## Abstract

This study aimed to assess and analyze the morphology and dimensions of the nasopalatine canal (NPC), as well as the adjacent buccal osseous plate (BOP), and to investigate the effect of gender, edentulism, NPC types, absence of maxillary central incisors (ACI) and age on the NPC and BOP, using cone beam computed tomography (CBCT). A total of 124 CBCT examinations (67 female and 57 male patients) were retrospectively included and evaluated. The assessment of the dimensions of the NPC, as well as the dimensions of the adjacent BOP, was performed by three Oral and Maxillofacial Radiologists on reconstructed sagittal and coronal CBCT sections under standardized conditions. Regarding the dimensions of the NPC and the adjacent BOP, the mean values were significantly higher among males than females. Furthermore, edentulous patients showed a significant reduction in BOP dimensions. Additionally, NPC types showed a significant effect on the length of the NPC, and the ACI had a significant effect on reducing BOP dimensions. Age had a significant effect on the diameter of the incisive foramen, with the mean values generally increasing with an increasing age. CBCT imaging of this anatomical structure contributes significantly to its full assessment.

## 1. Introduction

The nasopalatine canal (NPC) is an important anatomical structure of the maxilla located along the middle of maxilla, in between and slightly posterior to the central maxillary incisors [1]. The NPC opens in the nasal cavity with the Stenson foramen (SF) and extents to the oral cavity through the incisive foramen (IF) [2,3,4]. The NPC carries the nasopalatine nerve, the terminal branch of the descending nasopalatine artery, as well as small salivary glands, fat and fibrous connective tissue [5,6]. Branches of the maxillary artery and trigeminal nerve may also pass through the NPC [5,7,8] and provide innervation and vascularization to the anterior palate from canine to canine [9].

This particular anatomical region is also considered important both in terms of the aesthetic rehabilitation of the anterior maxilla, and because this area is often involved in cases of dental trauma and tooth loss. Notably, oftentimes the rehabilitation of this region mandates implant placement [5,10,11,12] and thus, the evaluation of the NPC and its variations (via imaging) plays an important role in the prevention of complications during implant surgery, such as bleeding, pain, sensory dysfunction and potential failures in the osseointegration process [5,8,11,12,13]. Similarly, preoperative evaluation of the NPC, as well as the adjacent buccal osseous plate (BOP) is particularly important for local anesthesia of maxillary anterior teeth [5,14,15,16], maxillary surgery, enucleation of nasopalatine duct cysts, surgical extraction of impacted teeth or mesiodens, apical resection of central incisors, rapid palatal enlargement with surgical support and Le Fort I osteotomy [2,17,18].

Diagnostic imaging of the anterior maxilla includes two dimensional (2D) techniques, such as panoramic tomography and intra-oral radiography [8], as well as spiral tomography and/or multislice computed tomography (MSCT) [19]. Although the NPC can be visualized on 2D radiographic images [7,8,15,20], it is generally agreed that the inherent limitations of these techniques such as magnification, overlap and distortion may affect the accurate assessment of its anatomic variations [13,20]. The advent of cone-beam computed tomography (CBCT) expanded the boundaries of imaging, from two-dimensional (2D) to three-dimensional (3D), also providing new diagnostic perspectives in dentistry [6]. Hence, CBCT has already established itself as the volumetric 3D imaging modality of choice among general dental practitioners and specialists (e.g., oral and maxillofacial surgeons, orthodontists, etc.) as this has, primarily, a relative low radiation dose and good image quality [21,22,23,24,25,26,27].

This retrospective study specifically aims (a) to assess and analyze the morphology and dimensions of the NPC, as well as the adjacent BOP, and (b) to investigate the effect of gender, edentulism, NPC types, absence of maxillary central incisors (ACI) and age on the NPC and BOP, using CBCT.

## 2. Materials and Methods

### 2.1. Ethical Approval

The CBCT scans used in this study were obtained and coded from the archives of the Department of Oral Diagnosis and Radiology, School of Dentistry, National and Kapodistrian University of Athens, Greece.

Additionally, the study was approved by the institutional research ethics committee (protocol number 26800 and registration code 568, Athens, Greece) being also in accordance with the Declaration of Helsinki.

### 2.2. Study Material

From a series of 325 randomly selected full-volume [field of view (FOV) 15 × 12 cm] CBCT consecutive scans of solely Greek patients (other ethnic groups were not included), a number of 124 scans [male and female patients, dentate and edentulous, aged 13 to 82 years (mean age 48.73 years/SD 18.73 years) were included in this study. CBCT scans with sufficient sharpness and contrast for adequate visualization and assessment of osseous structures, such as NPC and BOP, were considered eligible. In contrast, CBCT scans of patients with jaw fractures, the presence of residual roots, nasopalatine pathology (e.g., nasopalatine duct cysts), root remnants, and bone grafts in the anterior maxilla, were excluded. Additionally, scans suffering from poor quality, as well as presence of artifacts, related to the region of interest, were also excluded.

These 124 CBCT scans were retrospectively evaluated for NPC morphology and anatomic variations, as well as the adjacent BOP. The scans were required for a variety of reasons (e.g., preoperative implant planning, orthodontic and/or orthognathic evaluation, examination for the presence of supernumerary teeth and/or impaction, etc.). Gender, edentulism, NPC types, ACI and age were simultaneously recorded.

### 2.3. Imaging Procedure

All CBCT scans were obtained using a New Tom VGi CBCT imaging unit [(Cefla, Bologna, Italy) (operating parameters: 3.66 mA, 110 kV, voxel size 0.3 mm, FOV 15 × 12 cm and exposure time 3.6 s)].

Reconstructed sagittal and coronal CBCT sections (1 mm slice thickness) were used for the assessment of anatomical structures of interest. For their ideal depiction, the reconstructed sagittal slice was orientated perpendicular to the floor of the nasal cavity and hard palate antero-posteriorly, being also parallel to the course of the NPC.

### 2.4. Image Evaluation

The image evaluation and relevant measurements were performed by three Oral and Maxillofacial Radiologists (OMFRs) independently on reconstructed sagittal and coronal CBCT sections under standardized conditions [6], in 4 viewing sessions of 31 scans during one month. Four weeks after the first assessment, all scans were reassessed by one OMFR, to assess intra-observer coefficients. Reconstructed CBCT images were analyzed and evaluated using a tower workstation (Hp Z240, HP Inc., Palo Alto, California, CA, USA) and a 21.3″ FlexScan EIZO MX210 monitor (Eizo Nanao Corporation, Ishikawa, Japan) with a resolution of 1600 × 1200 pixels.

The required measurements (in mm) were: (#1) the diameter of IF, (#2) the diameter of SF, (#3) the diameter in the middle of the NPC, (#4) the total length of the NPC, (#5) the crestal distance from the buccal border of the IF to the facial aspect of the BOP, (#6) the distance midway from the buccal bone wall of the NPC to the facial aspect of the bone wall, using a horizontal line from the palatal border of the IF, and (#7) the most cranial distance from the buccal border in the middle of the NPC to the facial aspect of the buccal bone wall. These were measured on the reconstructed sagittal CBCT images (Figure 1 and Figure 2). All measurements were performed using the manufacturer’s specialized computer software (NNT v.6.2, Verona, Italy) [6,27,28]. Anatomical types of NPC were also recorded and classified as: a single canal (A), two parallel canals (B), variations of the Y-type of canal with one IF and two or more SF (C). This was performed on the reconstructed coronal CBCT images (Figure 3) [6,27,28].

### 2.5. Statistical Analysis

Data were analyzed using IBM-SPSS v.22 and Minitab v.16.1. Anatomical variants of the NPC were assessed using MS Excel 2013 v.15.0 and Statistica v.10 Enterprise.

Cohen’s Kappa, Fleiss’ Kappa, and intra class correlation (ICC) tests were used to assess observational quality. Data were subjected to analysis of variance (ANOVA one-way), *t*-test and Pearson’s correlation test and analyzed using IBM-SPSS v.22 and Minitab v.16.1. The level of significance was set at *p* ≤ 0.05.

## 3. Results

Out of the selected 124 CBCT scans, 57 (46%) were male and 67 (54%) were female patients, both dentulous (93.5%) and edentulous (6.5%). Regarding ACI, 17.7% of the sample were found to have 0 central incisors, 9.7% had only one and 72.6% had two central incisors.

### 3.1. Descriptive Analysis of the NPC and the Adjacent BOP

Evaluation of the different anatomic types of NPC resulted in the detection of a single canal in 56.5%, two separate parallel canals in 25%, and variations of the Y-type canal in 18.5% of the scans, respectively (Figure 3). The descriptive analysis of NPC dimensional evaluation parameters is presented in Table 1.

Intra-observer coefficients were calculated for the first observer only (Cohen’s Kappa > 0.95 and ICC > 0.95), suggesting excellent intra-observer agreement (*p* < 0.01). Inter-observer coefficients were calculated between the three OMFRs (Fleiss Kappa = 0.83 and ICC > 0.95), suggesting excellent inter-observer agreement *(p* < 0.01).

### 3.2. Analysis of Gender, Edentulism, NPC Types, ACI and Age Affecting NPC and Adjacent BOP

Table 2 clearly shows that mean values were found to be significantly higher in males than females (*p* < 0.05) [Measurement: #2 (*p* = 0.001); #3 (*p* = 0.033); #4 (*p* = 0.003); #5 (*p* = 0.024); #6 (*p* = 0.000); #7 (*p* = 0.029)]. Interestingly, edentulous patients showed a significant reduction in BOP dimensions [Measurement: #5 (*p* = 0.000); #6 (*p* = 0.000)]. Additionally, NPC types had a significant effect on SF diameter [Measurement #2 (*p* = 0.009)] and mid-NPC diameter [Measurement #3 (*p* = 0.040)]. It is worth noting that mean values were generally higher for the type C variant. Regarding the total length of the NPC, this showed higher mean values in the type A variant of the NPC [Measurement #4 (*p* = 0.000)]. Additionally, ACI had a significant effect on the reduction in BOP dimensions, as well as on the total length of the NPC. Moreover, the width of the adjacent BOP gradually decreased depending on the presence or absence of the central incisors [Measurements: #5 (*p* = 0.000), #6 (*p* = 0.000)]. Accordingly, the mean values of the total length of the NPC decreased significantly [Measurement #4 (*p* = 0.046)] (Table 2). Furthermore, patients’ age had a significant effect on the IF diameter, as the mean values generally increased with increasing age [Measurement #1 (*p* = 0.016)] (Table 3).

## 4. Discussion

The present study stands as a morphological analysis of the NPC, as well as its anatomic variations, using CBCT. Similarly, the adjacent BOP was co-evaluated. Our investigation on the effect of gender, edentulism, NPC types, ACI and age on NPC and BOP showed statistically significant results.

Assessment of the NPC has been carried out in previous studies, using Multislice CT (MSCT) scans [5,8,16], hi-resolution magnetic resonance imaging (MRI) [1], micro-CT images [14] and CBCT imaging [6,11,27,28,29,30,31]. In our study, CBCT imaging was used to assess and analyze the morphology and dimensions of the NPC, and its anatomic variations, as well as its adjacent BOP. It is worth noting that CBCT has a lower radiation dose than MSCT [6]. Given that CBCT has comparable and/or higher spatial resolution to that of MSCT, it is considered suitable for imaging and assessing various subtle anatomical structures such as that of the NPC and its adjacent BOP. Additionally, due to its inherent limitations, 2D imaging/radiography struggles to provide accurate information about the anatomical structures being assessed [8,22,27].

Furthermore, the specific software provided by the manufacturer of the CBCT unit used in the present study allowed us to make accurate measurements of the structures under investigation. Specifically: (a) the mean diameter of the IF was found to be 6.00 mm, (b) that of the SF was 3.10 mm, (c) that at the middle of the NPC was 2.04 mm, (d) that of the total length of the NPC was 12.16 mm, (e) that of the crestal distance from the buccal border of the IF to the facial aspect of the BOP was 6.86 mm, (f) the distance midway from the buccal bone wall of the NPC to the facial aspect of the bony wall (using a horizontal line from the palatal border of the IF) was 6.86 mm, and (g) the most cranial distance from the buccal border (in the middle of the NPC) to the facial aspect of the buccal bone wall was 7.61 mm. Our findings, regarding the dimensions of the NPC and its adjacent BOP, are consistent with those reported by Bornstein et al. (2011) [6]; on the other hand, they added more detailed information about the anatomical description of the NPC and its adjacent BOP. This fact increases the validity of our study.

In the same vein, the standardized projection protocol we used allowed us to also assess the different anatomic types of the NPC. In this manner, a single canal was detected in 56.5%, two separate parallel canals in 25%, and Y-type variations of the canal were observed in 18.5% of the evaluated CBCT scans. Our findings were in accordance with those of other studies [6,14,27], while Bornstein et al. (2011) reported that in the most cases a single canal was present [6,14]. In addition, NPC types had a significant effect on SF and mid-NPC diameters [measurement #3 (*p* = 0.040)], with mean values generally higher for the type C variant. The total length of the NPC showed higher mean values for the type A NPC. In the Bornstein et al. (2011) study of 100 subjects using CBCT, it was reported that the type of NPC had a significant effect only on IF diameter, with mean values highest for type A, followed by type B and C variants in a descending order [6].

Moreover, the results of our study showed that ACI had a significant effect on the reduction in BOP dimensions, which is particularly evident in edentulous patients. These findings are consistent with those of other studies [6,16,27]. However, a limitation of our study was that time elapsed, since the loss of central incisors was not known. Nevertheless, a comparative advantage compared to other studies was that edentulism of the maxilla was studied as a separate variable.

Regarding gender preference in NPC and adjacent BOP dimensions, we found that mean values of these measures—except that of IF—were significantly higher among males than females. Remarkably, analogous findings were also reported in a recent anatomical study of 1000 patients using CBCT [27]. It is worth noting that similar findings have been reported in other studies [6,16]. Thus, it is clear that gender significantly influences the dimensions of the NPC and its adjacent BOP, with male patients generally showing higher mean values.

As for the effect of age, similar studies using CBCT scans divided patients according to age into different groups [27,28,31], while in other CBCT studies, such as our study, patients were not divided into age groups [6,29]. Moreover, the results of our study showed that age had a significant effect on IF diameter, with mean values generally increasing with increasing age. In contrast, Bornstein et al. (2011) observed that age had a significant effect on NPC length, with mean values generally decreasing with increasing age [6].

Clinically, the restoration of the anterior esthetic zone often involves implant placement. This makes NPC assessment crucial [7,8,9,32]. It is worth noting that the enucleation of the NPC structure (nasopalatine nerve and artery, fibrous connective tissue and fat, as well as small salivary glands), application of bone grafting, and insertion of an implant directly into the NPC have been applied in numerous studies to rehabilitate severe atrophy of the maxilla [9,16,29,33,34,35]. Hence, understanding the anatomical characteristics of the NPC and its role in implant placement is mandatory. In this context, intra- and post-operative complications, such as bleeding, sensory deficit, failure of osseointegration and nasopalatine duct cyst formation can be prevented [29,35,36]. Based upon all of our aforementioned findings, a thorough radiological analysis is mandatory when planning the insertion of a dental implant in the anterior maxilla [6]. Hence, the results of our study contribute substantially to preoperative planning for implant placement in this subtle and delicate anatomical region.

## 5. Conclusions

Utilizing CBCT imaging in our study enabled us to thoroughly study and analyze NPC and its anatomical characteristics. We showed that CBCT imaging of the NPC provides the clinician with critical information for evaluating implant placement in the maxillary esthetic zone. Furthermore, the results of our study showed that the NPC, as well as its adjacent BOP, were affected by gender, edentulism, NPC types, ACI and age. Thorough morphologic evaluation of the NPC, preferably using CBCT, is beneficial to avoid intra- and post-operative complications during anterior maxillary surgeries. Future anatomical and morphological studies, which may use larger cohorts as well as ethnic differences, may further contribute to the understanding and surgical approach of the anterior maxilla.

## Figures and Tables

**Figure 1 diagnostics-13-01787-f001:**
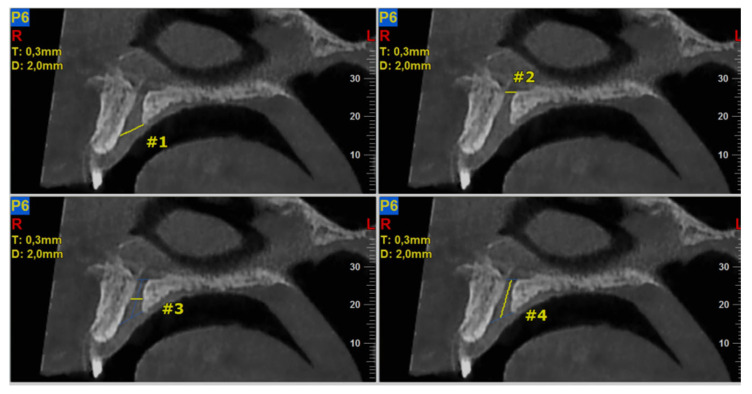
The following landmarks were selected for nasopalatine canal (NPC) analysis of the sagittal cone beam computed tomography (CBCT) images (all measurements in mm): #1—the diameter of IF, #2—the diameter of SF, #3—the diameter in the middle of the NPC, #4—the total length of the NPC.

**Figure 2 diagnostics-13-01787-f002:**
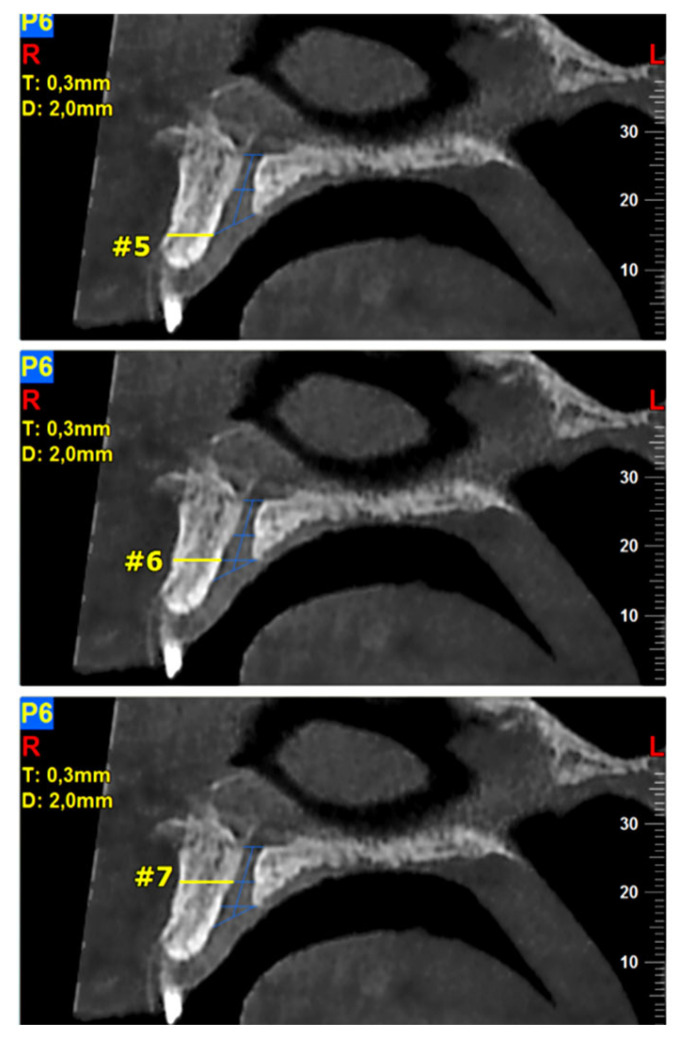
The following landmarks were selected for the buccal osseous plate (BOP) analysis of the sagittal cone beam computed tomography (CBCT) images (all measurements in mm): #5—the crestal distance from the buccal border of the incisive foramen (IF) to the facial aspect of the BOP, #6—the distance midway from the buccal bone wall of the NPC to the facial aspect of the bone wall, using a horizontal line from the palatal border of the IF, #7—the most cranial distance from the buccal border in the middle of the NPC to the facial aspect of the buccal bone wall.

**Figure 3 diagnostics-13-01787-f003:**
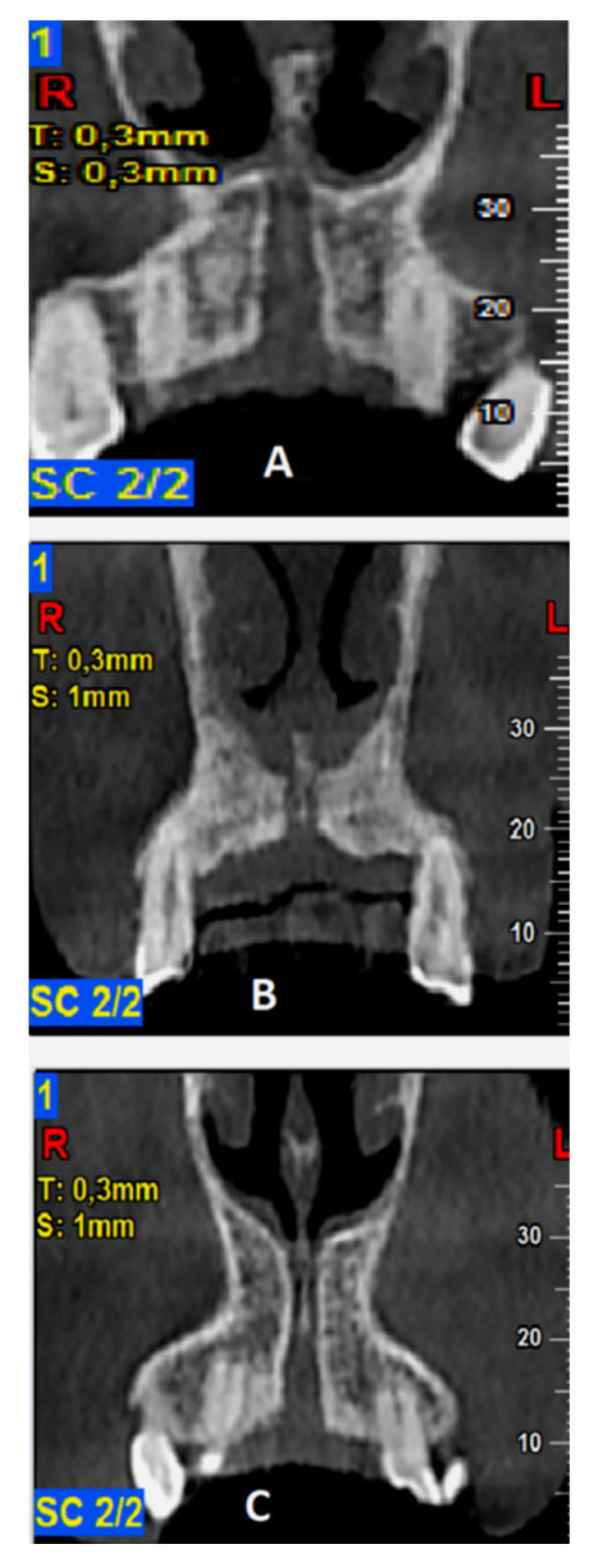
Classification of anatomic types of the nasopalatine canal (NPC) as evaluated in coronal cone beam computed tomography (CBCT) images: (**A**)—a single canal, (**B**)—two parallel canals, (**C**)—variations of the Y-type of canal with one incisive foramen (IF) and two or more Stenson foramen (SF).

**Table 1 diagnostics-13-01787-t001:** Analysis of the dimensions (in mm) of the nasopalatine canal (NPC) and the adjacent buccal osseous plate (BOP) using reconstructed sagittal sections from cone beam computed tomography (CBCT) images.

Measurements	*n*	Mean	SD	SEM	95% CI	Min	Max
#1	124	6.00	1.36	0.12	5.76–6.24	2.87	11.43
#2	124	3.10	1.29	0.11	2.86–3.32	0.70	7.73
#3	124	2.04	1.01	0.09	1.86–2.21	0.43	5.67
#4	124	12.16	2.95	0.26	11.63–12.68	5.27	21.40
#5	124	6.86	1.66	0.14	6.56–7.15	0.90	11.33
#6	124	6.86	1.74	0.15	6.55–7.17	1.17	11.33
#7	124	7.61	1.66	0.14	7.31–7.90	3.13	11.13

#1–#7: Measurements that correspond to the distances shown on Figure 1 and Figure 2; *n*: total sample number; SD: standard deviation; SEM: standard error of the mean; CI: confidence interval (95%) for mean.

**Table 2 diagnostics-13-01787-t002:** The effect of gender, edentulism, nasopalatine canal (NPC) types and absence of maxillary central incisors (ACI) on the dimensions (in mm) of the NPC and the adjacent buccal osseous plate (BOP) using reconstructed sagittal sections from cone beam computed tomography (CBCT) images.

			Measurements (in mm)						
		*n*	#1	#2	#3	#4	#5	#6	#7
			Mean (SD)	Mean (SD)	Mean (SD)	Mean (SD)	Mean (SD)	Mean (SD)	Mean (SD)
Variables	General	124	6.00 (1.36)	3.10 (1.29)	2.04 (1.01)	12.16 (2.95)	6.86 (1.66)	6.86 (1.74)	7.61 (1.66)
Gender	Male	57	6.14 (1.37)	3.51 (1.19)	2.25 (1.07)	13.01 (2.96)	7.23 (1.56)	7.45 (1.57)	7.96 (1.64)
	Female	67	5.89 (1.35)	2.74 (1.27)	1.86 (0.93)	11.44 (2.75)	6.56 (1.69)	6.37 (1.73)	7.31 (1.64)
	*p*-value (*t*-test)	-	0.304	0.001 *	0.033 *	0.003 *	0.024 *	0.000 *	0.029 *
Edentulism	Yes	8	5.83 (1.69)	3.85 (1.84)	1.95 (0.83)	10.56 (2.42)	3.68 (2.14)	4.39 (2.24)	6.84 (1.45)
	No	116	6.02 (1.34)	3.04 (1.24)	2.05 (1.02)	12.27 (2.96)	7.08 (1.38)	7.03 (1.57)	7.66 (1.67)
	*p*-value (*t*-test)	-	0.714	0.086	0.787	0.113	0.000 *	0.000 *	0.177
NPC types	A	70	5.88 (1.22)	2.83 (1.35)	1.84 (0.98)	13.09 (2.94)	6.91 (1.68)	6.82 (1.79)	7.66 (1.69)
	B	31	5.83 (1.29)	3.22 (1.06)	2.28 (1.12)	10.53 (2.50)	6.75 (1.50)	6.95 (1.62)	7.50 (1.61)
	C	23	6.62 (1.70)	3.74 (1.18)	2.33 (0.80)	11.52 (2.44)	6.83 (1.86)	6.86 (1.78)	7.58 (1.72)
	*p*-value (One-way ANOVA)	-	0.054	0.009 *	0.040 *	0.000 *	0.929	0.944	0.900
ACI	0	22	5.52 (1.25)	3.37 (1.58)	1.90 (1.01)	10.85 (3.21)	5.25 (1.95)	5.58 (1.81)	7.20 (1.47)
	1	12	5.67 (0.95)	2.74 (1.38)	1.53 (0.72)	11.74 (2.14)	6.25 (1.64)	6.68 (2.21)	7.95 (2.29)
	2	90	6.17 (1.41)	3.08 (1.20)	2.14 (1.03)	12.54 (2.90)	7.34 (1.29)	7.20 (1.51)	7.66 (1.61)
	*p*-value (One-way ANOVA)	-	0.088	0.377	0.115	0.046 *	0.000 *	0.000 *	0.390

#1–#7: Measurements that correspond to the distances shown on Figure 1 and Figure 2; *n*: total number of the sample; SD: standard deviation; * Statistically significant at level *p* ≤ 0.05.

**Table 3 diagnostics-13-01787-t003:** The effect of age on the dimensions (in mm) of the nasopalatine canal (NPC) and the adjacent buccal osseous plate (BOP) using reconstructed sagittal sections from cone beam computed tomography (CBCT)images.

	*n*	Mean	SD	Minimum	Maximum	Coefficient	*p*-Value (Pearson’s Correlation Test)
Age (in years)	124	48.73	18.73	13	83	-	-
Measurements (in mm)						-	-
#1	124	6.06	1.43	3.00	11.50	0.216	0.016 *
#2	124	3.25	1.35	0.30	7.80	0.174	0.053
#3	124	2.04	1.13	0.30	5.80	0.109	0.230
#4	124	12.27	2.97	5.40	21.30	−0.053	0.561
#5	124	6.73	1.75	0.90	11.40	−0.066	0.464
#6	124	6.68	1.84	0.60	11.40	−0.004	0.965
#7	124	7.38	1.80	2.40	11.70	−0.048	0.593

#1–#7: Measurements that correspond to the distances shown on Figure 1 and Figure 2; *n*: total number of the sample; SD: standard deviation; * Statistically significant at level *p* ≤ 0.05.

## Data Availability

The datasets used and/or analyzed during the current study are available from the corresponding author upon reasonable request.

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
