# Peer review of "Morphological Assessment of Nasopalatine Canal Using Cone Beam Computed Tomography: A Retrospective Study of 124 Consecutive Patients"

_diagnostics, 2023, doi:10.3390/diagnostics13101787_

Round 1

Reviewer 1 Report

Dear authors,

I congratulate you on your interesting and well-conducted work. The various sections of the article are correctly described, and the research theme well fits with that of the special issue. I only ask you to carefully review the text regarding the language.

Additional comments

1. What is the main question addressed by the research?

- The main question addressed by the research is just merely descriptive of the anatomy of the nasopalatine area.

2. Do you consider the topic original or relevant in the field? Does it address a specific gap in the field?

- An anatomical description can obviously never be entirely original. The present paper nevertheless adds interesting details to the topic.

3. What does it add to the subject area compared with other published material?

- It adds original measurements on a quite large sample

4. What specific improvements should the authors consider regarding the methodology? What further controls should be considered?

- None, the analysis of the nasopalatine region is accurate and complete

5. Are the conclusions consistent with the evidence and arguments presented and do they address the main question posed?

- An anatomical work is descriptive, so the conclusion section is, in this case, less relevant than in other types of issues

6. Are the references appropriate?

- Yes.

7. Please include any additional comments on the tables and figures.

- Improve the quality of the images, if possible

My best regards.

No serious or formal errors are found, but only the need to make the language more fluid in some complex passages. Native speaker revision may be useful.

Author Response

Response to Reviewer 1 Comments

Point 0: I congratulate you on your interesting and well-conducted work. The various sections of the article are correctly described, and the research theme well fits with that of the special issue. I only ask you to carefully review the text regarding the language.

Response 0: We thank you very much for your positive opinion. We made structural and phraseological changes to the text in order to make the language more fluid in some complex passages, according to your comments and suggestions.

Point 1: What is the main question addressed by the research? - The main question addressed by the research is just merely descriptive of the anatomy of the nasopalatine area.

Response 1: Thank you very much for this comment. Indeed, the main question addressed by our research was just merely to describe the anatomy of the nasopalatine area to explore anatomical variations of the nasopalatine canal and its adjacent buccal osseous plate.

Point 2: Do you consider the topic original or relevant in the field? Does it address a specific gap in the field? - An anatomical description can obviously never be entirely original. The present paper nevertheless adds interesting details to the topic.

Response 2: Thank you again for your positive comment.

Point 3: What does it add to the subject area compared with other published material? - It adds original measurements on a quite large sample.

Response 3: Thank you very much for your positive opinion. This contributes to a better knowledge of the nasopalatine canal variants.

Point 4: What specific improvements should the authors consider regarding the methodology? What further controls should be considered? - None, the analysis of the nasopalatine region is accurate and complete.

Response 4: We thank you a lot for your positive comment.

Point 5: Are the conclusions consistent with the evidence and arguments presented and do they address the main question posed? - An anatomical work is descriptive, so the conclusion section is, in this case, less relevant than in other types of issues.

Response 5: Thank you very much for this comment, which finds us in complete agreement. Actually, due to the fact that an anatomical work is descriptive, the conclusion section is less relevant than in other types of issues.

Point 6: Are the references appropriate? - Yes.

Response 6: Thank you again for your positive opinion.

Point 7: Please include any additional comments on the tables and figures. - Improve the quality of the images, if possible.

Response 7: Thank you a lot for this insightful comment. We have improved the quality of the images, according your suggestion.

Reviewer 2 Report

Dear authors, I would like to congratulate you for conducting the present study. Here goes a few concerns and suggestions:

Please be aware that the affiliation requires a superscript number after each author. Please check the guidelines.

I would suggest the authors to increase the number of keywords to 5 and to place them by alphabetic order.

Regarding the methods, how were the examinations selected? All consecutive according to the title. That information should be present in the methods.

Why were the CBCT examinations performed? Not due to the present research, right?

Where were the patients from? All Greek?

Any specific ethnic groups?

Was this a convenience sample?

How many cases were excluded? What was the exclusion rate? And why were they excluded?

What was the CBCT voxel size?

Did the authors conducted any intra- and inter-observer reliability tests?

May the authors debate the study strength in the Discussion?

May the authors debate the study limitations?

May the authors debate the possible generalization of the results?

May the authors debate the internal and external validity of the study?

May the authors give recommendations for future research?

Minor grammar and syntax fixing are needed

Author Response

Response to Reviewer 2 Comments

Point 0: Dear authors, I would like to congratulate you for conducting the present study.

Response 0: We thank you very much for your positive opinion.

Point 1: Please be aware that the affiliation requires a superscript number after each author. Please check the guidelines.

Response 1: Thank you very much for this observation. We have omitted the superscript number after each author because all authors belong to the same department. Nevertheless, we have modified the text according guidelines (Author List and Affiliations Section; Page 1).

Point 2: I would suggest the authors to increase the number of keywords to 5 and to place them by alphabetic order.

Response 2: Thank you so much for this valuable suggestion. We have increased the number of keywords to 5, placing them by alphabetic order, according your suggestion (Keywords Section; Page 1).

Point 3: Regarding the methods, how were the examinations selected? All consecutive according to the title. That information should be present in the methods.

Response 3: We thank you so much for this helpful and insightful comment. Following your suggestion we have incorporated that information in the Materials and Methods Section (Materials and Methods Section; 2.2 Study Material; Paragraph 1; Page 2).

Point 4: Why were the CBCT examinations performed? Not due to the present research, right?

Response 4: Thank you for this comment. The scans were required for a variety of reasons (e.g., preoperative implant planning, orthodontic and/or orthognathic evaluation, examination for the presence of supernumerary teeth and/or impaction, etc.). This was already presented in Materials and Methods Section (Materials and Methods Section; 2.2 Study Material; Paragraph 2; Page 2).

Point 5: Where were the patients from? All Greek?

Response 5: Thank you so much for this insightful comment. In this retrospective study, archived CBCT scans of Greek patients were examined. This is now presented in the revised manuscript. (Materials and Methods Section; 2.2 Study Material; Paragraph 1; Page 2).

Point 6: Any specific ethnic groups?

Response 6: Thank you again for this comment. The study material was consisted of Greek patients with no specific ethnic groups. This is now presented in the revised manuscript, according your suggestion. We believe that is now brief and clear to the readers (Materials and Methods Section; 2.2 Study Material; Paragraph 1; Page 2).

Point 7: Was this a convenience sample?

Response 7: Thank you for this comment. We agree that our study material was a convenience sample, due to geographical proximity. As mentioned above, all patients were Greek with no specific ethnic groups.

Point 8: How many cases were excluded? What was the exclusion rate? And why were they excluded?

Response 8: We thank you so much for this comment. A number of 201 scans were excluded in this study (exclusion rate: 61.85%). From a series of 325 randomly selected CBCT consecutive scans, a number of 124 scans were included in this study. CBCT scans with sufficient sharpness and contrast for adequate visualization and assessment of osseous structures, such as NPC and BOP, were considered eligible. In contrast, CBCT scans of patients with jaw fracture, presence of residual roots, nasopalatine pathology (e.g., nasopalatine duct cyst), root remnant, bone graft in the anterior maxilla, were excluded. Additionally, scans suffering from poor quality, as well as presence of artifacts, related to the region of interest, were also excluded. This was already presented in Materials and Methods Section (Materials and Methods Section; 2.2 Study Material; Paragraph 1; Page 2).

Point 9: What was the CBCT voxel size?

Response 9: We thank you for this insightful comment. The CBCT voxel size was 0.3mm. This is now presented in the revised manuscript, according your suggestion in the Materials and Methods Section (Materials and Methods Section; 2.3 Imaging Procedure; Paragraph 1; Page 2).

Point 10: Did the authors conducted any intra- and inter-observer reliability tests?

Response 10: Thank you for this comment.

The image evaluation and relevant measurements were performed by three Oral and Maxillofacial Radiologists (OMFRs) independently on reconstructed sagittal and coronal CBCT sections under standardized conditions, in 4 viewing sessions of 31 scans during one month. Four weeks after the first assessment, all scans were reassessed by one OMFR, to assess intra-observer coefficients. This was already presented in Materials and Methods Section (Materials and Methods Section; 2.4 Images Evaluation; Paragraph 1; Page 3).

Intra-observer coefficients were calculated for the first observer only (Cohen’s Kappa >0.95 and ICC >0.95), suggesting excellent intra-observer agreement (P <0.01). Inter-observer coefficients were calculated between the three OMFRs (Fleiss Kappa = 0.83 and ICC >0.95), suggesting excellent inter-observer agreement (P <0.01). This was already presented in Results Section (Results Section; 3.1. Descriptive Analysis of the NPC and the adjacent BOP; Paragraph 2; Page 5).

Point 11: May the authors debate the study strength in the Discussion?

Response 11: Thank you so much for this helpful comment. We have enriched the Discussion, according your suggestion, in order to debate the strength of our study. This is now presented in Discussion Section (Discussion Section, Paragraph 3, Page 7) (Discussion Section, Paragraph 8, Page 8).

Nevertheless, a comparative advantage compared to other studies was that edentulism of maxilla was studied as a separate variable. This was already presented in Discussion Section (Discussion Section, Paragraph 5, Page 7).

Point 12: May the authors debate the study limitations?

Response 12: Thank you for this comment. A limitation of our study was that the time elapsed since the loss of central incisors was not known. This was already presented in Discussion Section (Discussion Section, Paragraph 5, Page 7).

Point 13: May the authors debate the possible generalization of the results?

Response 13: We thank you very much for this comment. The results of this study indicate that gender significantly influences the oro-facial dimensions of the NPC and its adjacent BOP, with male patients generally exhibiting higher mean values. This is now presented in Discussion Section (Discussion Section, Paragraph 6, Page 7).

Point 14: May the authors debate the internal and external validity of the study?

Response 14: Thank you so much for this comment.

The image evaluation and relevant measurements were performed by three Oral and Maxillofacial Radiologists (OMFRs) independently on reconstructed sagittal and coronal CBCT sections under standardized conditions, in 4 viewing sessions of 31 scans during one month. Four weeks after the first assessment, all scans were reassessed by one OMFR, to assess intra-observer coefficients. This was already presented in Materials and Methods Section (Materials and Methods Section; 2.4 Images Evaluation; Paragraph 1; Page 3). Furthermore, Intra-observer coefficients were calculated for the first observer only (Cohen’s Kappa >0.95 and ICC >0.95), suggesting excellent intra-observer agreement (P <0.01). Inter-observer coefficients were calculated between the three OMFRs (Fleiss Kappa = 0.83 and ICC >0.95), suggesting excellent inter-observer agreement (P <0.01). This was already presented in Results Section (Results Section; 3.1. Descriptive Analysis of the NPC and the adjacent BOP; Paragraph 2; Page 5). Thus, the internal validity of the study was increased.

In our study, we used random sampling to obtain a representative sample from the population we are studying. So, from a series of 325 randomly selected CBCT consecutive scans, a number of 124 scans [male and female patients, dentate and edentulous, aged 13 to 82 years (mean age 48.73 years/SD 18.73 years)] were included in this study. This was already presented in Materials and Methods Section (Materials and Methods Section; 2.2 Study Material; Paragraph 1; Page 2). Therefore, the external validity of this study was increased.

Point 15: May the authors give recommendations for future research?

Response 15: Thank you for this comment. Following your suggestion we have incorporated that information in the Conclusions Section (Conclusions Section, Paragraph 1, Page 8).

Round 2

Reviewer 2 Report

Dear author, I have no more concerns.